# Mechanisms Underlying the Virulence Regulation of *Vibrio alginolyticus* ND-01 *pstS* and *pstB* with a Transcriptomic Analysis

**DOI:** 10.3390/microorganisms10112093

**Published:** 2022-10-22

**Authors:** Xin Yi, Xiaojin Xu, Xin Qi, Yunong Chen, Zhiqin Zhu, Genhuang Xu, Huiyao Li, Emma-Katharine Kraco, Haoyang Shen, Mao Lin, Jiang Zheng, Yingxue Qin, Xinglong Jiang

**Affiliations:** 1Key Laboratory of Healthy Mariculture for the East China Sea, Engineering Research Center of the Modern Technology for Eel Industry, Fisheries College, Jimei University, Xiamen 361021, China; 2Fisheries Research Institute of Fujian, Xiamen 361013, China; 3School of Freshwater Sciences, University of Wisconsin-Milwaukee, 600 East Greenfield Avenue, Milwaukee, WI 53204, USA

**Keywords:** *pstS*, *pstB*, *Vibrio alginolyticus*, virulence, homology analysis, RNAi, transcriptome

## Abstract

*Vibrio alginolyticus* is a common opportunistic pathogen of fish, shrimp, and shellfish, and many diseases it causes can result in severe economic losses in the aquaculture industry. Causing host disease was confirmed by several virulence factors of *V. alginolyticus.* To date, there have been no reports on the effect of the *pstS* gene on its virulence regulation of *V. alginolyticus*. The virulence mechanism of target genes regulating *V. alginolyticus* is worthy of further study. Previous studies found that *Fructus schisandrae* (30 mg/mL) inhibited the growth of *V. alginolyticus* ND-01 (OD_600_ = 0.5) for 4 h, while the expressions of *pstS* and *pstB* were significantly affected by *F**. schisandrae* stress. So, we speculated that *pstS* and *pstB* might be the virulence genes of *V. alginolyticus*, which were stably silenced by RNAi to construct the silencing strains *pstS*-RNAi and *pstB*-RNAi, respectively. After the expression of *pstS* or *pstB* gene was inhibited, the adhesion capacity and biofilm formation of *V. alginolyticus* were significantly down-regulated. The chemotaxis and biofilm formation ability of *pstS*-RNAi was reduced by 33.33% and 68.13% compared with the wild-type strain, respectively. Sequence alignment and homology analysis showed that *pstS* was highly conserved, which suggested that *pstS* played a vital role in the secretion system of *V. alginolyticus*. The *pstS*-RNAi with the highest silencing efficiency was selected for transcriptome sequencing. The Differentially Expressed Genes (DEGs) and GO terms were mapped to the reference genome of *V. alginolyticus*, including 1055 up-regulated genes and 1134 down-regulated genes. The functions of the DEGs were analyzed by GO and categorized into different enriched functional groups, such as ribosome synthesis, organelles, biosynthesis, pathogenesis, and secretion. These DEGs were then mapped to the reference KEGG pathways of *V. alginolyticus* and enriched in commonalities in the metabolic, ribosomal, and bacterial secretion pathways. Therefore, *pstS* and *pstB* could regulate the bacterial virulence of *V. alginolyticus* by affecting its adhesion, biofilm formation ability, and motility. Understanding the relationship between the expressions of *pstS* and *pstB* with bacterial virulence could provide new perspectives to prevent bacterial diseases.

## 1. Introduction

*Vibrio alginolyticus* is a conditional pathogenic bacterium affecting both cultured fish and invertebrates [1]. For example, the ‘‘milky disease’’ of mud crab, caused by pathogenic infection of bacteria *V*. *alginolyticus* in the last few years in Southern China, resulted in more than 60% of mortality and thus caused great economic losses [2]. At the same time, *V. alginolyticus* can infect eyes through wounds and cause endophthalmitis [3]. Causing host disease is determined by several virulence factors of *V. alginolyticus* under different conditions. Studies have shown that adhesion, extracellular products, an iron uptake system, and other virulence factors could affect the pathogenicity of *V. alginolyticus.* Adhesion was a critical virulence factor of pathogenic bacteria and a prerequisite for host bacterial infection [4]. The adhesion of *V. alginolyticus* to the surfaces of the host is important in regulating their interactions, as bacterial infections are linked to adhesion. To date, there have been no reports on the effect of the *pstS* gene of *V. alginolyticus* on its adhesion virulence regulation.

*F*. *schisandrae* is a medicinal plant in China, South Korea, and Japan. *F. schisandrae* has anti-inflammatory, detoxification, antioxidant, hepatoprotective, and antibacterial effects [5]. It has been elucidated that *F. schisandrae* extract can inhibit the growth of *Staphylococcus aureus* in vitro, and it can prevent and treat *S*. *aureus* in *Pelodiscus sinensis* infection [6]. In the early stage of our study, the antibacterial effect was detected by the Oxford cup method, using a Chinese medicine extract and its antibacterial activity against *V. alginolyticus* [7]. We found that the survival rate of *V. alginolyticus* decreased by 50%. The expression levels of *pstS* and *pstB* genes in the Pst system were significantly down-regulated. *F. schisandrae* had a noticeable antibacterial effect on *V. alginolyticus*. *F. schisandrae* and other compound Chinese medicines can effectively treat the *V. alginolyticus* disease of *Epinephelus coioides.* Although traditional Chinese medicine has a noticeable therapeutic effect on *V. alginolyticus*, its mechanism is still unclear [8].

The Pst system is closely related to bacterial virulence; they regulate bacterial adhesion biofilm formation and host invasion [9]. The *pstS* gene can significantly reduce the adhesion, cell invasion, bacterial motility, and biofilm formation ability of *Acinetobacter baumannii* [10]. In in vivo experiments, the minimum lethality of mice affected by the *pstS* mutant was significantly lower than that of wild-type *A**. baumannii*. *pstS* was related to bacterial virulence. Therefore, to clarify the virulence mechanism of *pst* mutant strains on the host, the *pst* gene of avian pathogenic *Escherichia coli* APEC O78:K80 strain 7122 was mutated by allele exchange technology, and the results showed that the virulence of *E. coli* in chickens was reduced [10]. It was found that causing the imbalance of bacterial phosphate transport and the change in bacterial surface composition was associated with the *pst* gene. In addition, in *pstS* and *pstC* genes, by using transposon technology, bacteria cannot cause a fatal infection [11]. The proteomics of *pst* mutant strains and wild-type *Edwardsiella*, but no transport protein of TSS3 and other virulence proteins, were found in the *pst* mutant strain. This may be the critical factor that the *pst* gene reduces bacterial virulence by altering the secretion of virulence proteins. This indicates that environmental degradation is the result of a multiplicity of causes, including aquaculture, increasing numbers of fishermen, harmful agricultural practices, and poor government policies and regulations [12].

Because of the pathogen and harm to cultured fishes, the pathogenic mechanism of *V. alginolyticus* has gained a great deal of interest among researchers. The pathogenicity of *V. alginolyticus* is considered to be associated with various virulence factors, and their roles as virulence factors still need to be identified. The aims of this study were as follows: (1) to uncover stable silent *pstS* and *pstB* by using RNAi technology; (2) to study the adhesion, biofilm formation, bacterial motility, and bacterial chemotaxis ability of *pstS* and *pstB* stable silencing strains of *V. alginolyticus*; and (3) to select the *pstS*-RNAi strain with the highest silencing efficiency for transcriptome sequencing. qRT-PCR was used to verify the transcriptome sequencing results and analyze the virulence mechanism of target genes regulating *V. alginolyticus*.

## 2. Materials and Methods

### 2.1. Bacterial Strains and Culture Conditions

*V. alginolyticus* (ND-01) was isolated from a diseased *E*. *coioides* and grown in LB broth or agar at 28 °C (pH = 7, 1% NaCl, 220 r.p.m.). The strain was confirmed as *V. alginolyticus* through 16S rDNA sequencing [13]. The Schistosome T cell-stimulating antigen (SM10) was purchased from China Tiangen Biological Co. Ltd. and preserved in our laboratory −80 °C refrigerator. The DH5α competent cells were purchased from Beijing All-style Gold Biological Co. Ltd. The SM10 and DH5α competent cells were cultured in LB broth at 37 °C (pH = 7, 1% NaCl, 220 r.p.m.). *E. coli* containing pACYC184 plasmid was stored in our laboratory and cultured in LB broth containing 1% (*w*/*v*) NaCl and chloramphenicol (34 ug/mL) at 37 °C. To verify the reliability of the previous transcriptome data, the wild strain was inoculated into LB liquid medium and cultured to a concentration of 10^8^ CFU/mL. *F. schisandrae* were added to the culture tubes at final concentrations of 30 mg/mL, wild strains were used as the control group, and total bacterial RNA of all samples was extracted and plated in triplicate.

### 2.2. Stable Gene Silencing

The methods used for constructing the stable silencing of the genes *V. alginolyticus* and treating *E. coli* SM10 according to the previous reports [14]. Among them, the pACYC184 vector was digested using restriction enzymes BamHI and SphI. The sequence of short hairpin (sh) RNA is shown in Table 1. The pACYC184 vectors were ligated using a T4 DNA ligase (TaKaRa).

A T4 DNA ligase (TaKaRa) was used to connect the pACYC184 vector and short hairpin (sh) RNA. The recombinant plasmid was transformed into *E. coli* containing DH5α plasmid. The conjugation experiment was carried out by transferring plasmids from *E. coli*. Recombinant *E. coli* containing SM10 was conjugated with *V. alginolyticus.* LB agar containing chloramphenicol (34 mg/mL) was used to select stable silenced *V. alginolyticus* [15].

### 2.3. DNA Isolation

The Wizard genomic DNA purification kit was used to extract the genomic DNA of *V**. alginolyticus* [16].

### 2.4. RNA Isolation

The total RNA of *V. alginolyticus* was extracted with TRIzol reagent (Transgenic Biotechnology, Beijing, China). Transscript^®^ All-in-One SuperMix for qPCR and gDNA Remover and RNase-free Water (Transgen Biotech, Beijing, China) were used for reverse transcription [17]. Synergy™ H1 (BioTek Instruments, Winooski, VT, USA) was used to measure the quality of the extracted RNA. The OD_260/280_ of ribonucleic acid was between 1.8 and 2.0, and the concentration was greater than 200 ng/μL. RNA was identified and purified by agarose gel electrophoresis.

### 2.5. qRT-PCR

The relative expression level of the target gene of *V. alginolyticus* was detected by real-time PCR. The SYBR-green method was performed on three controls and three stable silenced *V. alginolyticus* using TransStart Top Green qPCR SuperMix kit following the manufacturer’s instructions. The primer sequence is shown in Table 2. The reference gene is 16S rRNA [18]. The standard substance is recovered from each target gene by adding 1:10^−2^~1:10^−7^ crude extract dilutions to produce standard curves.

### 2.6. Growth Curve

A growth curve was drawn to detect the growth of the wild strain and stable silencing strain according to the previous description [19]. We adjusted the bacterial concentration to OD_600_ = 0.02, we take 10 μL bacterial liquid and 190 μL sterile LB liquid medium mixed in a 96-well plate, and each sample is set for eight parallel experiments. *V. alginolyticus* cultured overnight was incubated in a 28 °C biochemical incubator, we measure the OD_600_ measurements every hour until they achieve stable growth.

### 2.7. Soft Agar Plate Motility Assay

Bacterial motility on soft agar plates was determined according to the method described by [20]. The bacteria were cultured overnight in an LB medium, and the bacterial concentration was adjusted to OD_600_ = 0.2. Of the culture suspension, 1 μL was inoculated into an LB semi-solid agar plate (0.5% agar), and the plate was incubated at 28 °C for 16 h, the diameters of the bacterial colonies were measured in triplicate.

### 2.8. In Vitro Adhesion Assay

The mucus of *E**. coioides* was assayed with the previous method [21], and 20 μL mucus of *E.*
*coioides* was maintained on a glass slide (22 mm × 22 mm), supplemented with methanol for 20 min. The adhesion ability of the wild strain and stable silencing strain to the surface mucus of grouper was detected by incubation with bacterial suspension (10^8^ CFU/mL) at 28 °C for 2 h [22]. The experiment was carried out using both positive controls with *V. alginolyticus* and negative controls with sterile PBS.

### 2.9. Biofilm Formation Assay

Biofilm formation was assayed using the previous method [23], *V. alginolyticus* was cultured overnight for 12 h, and the OD_600_ was adjusted to 0.2 (2.0 × 10^8^ CFU/mL) with 0.01 M PBS (pH = 7.2). The suspension of 100 μL was added to the well of 96-well plate, each sample was repeated six times, and the 96-well plate was incubated at 28 °C for 24 h. The content in the 96-well plates was washed 3 times with 100 ul 0.01 M PBS (pH = 7.2) and was dried at 65 °C for 10 min. An amount of 125 mL 0.1% crystal violet solution (Merck KGaA, Darmstadt, Germany) was added to each well for 15 min. Finally, the dye biofilm was dissolved by using 200 uL 33% acetic acid, and a microplate reader (Bio-Rad, Hercules, CA, USA) was used to determine their OD_590_. The biofilms formed were measured as described above in triplicates.

### 2.10. Capillary Assay

Bacterial chemotaxis analyses were performed using a previously reported method [14]. A capillary tube with an inner diameter of 0.1 mm was filled with the mucus of *E. coioides*, leaving one end close. Then 300 μL bacterial suspension with OD_600_ = 1.0 (1.0 × 10^9^ CFU/mL) was loaded into test tube. The opening end of the capillary was placed in a syringe containing a bacterial solution. After incubation at 28 °C for 1 h, 100 μL capillary content was inoculated on LB agar plate to accurately determine the number of bacteria in the capillary tube. Chemotactic index = chemotactic cell of each strain/chemotactic cell of the wild-type strain [24].

### 2.11. Sequence Alignment and Homology Analysis

TRIzol reagent (Invitrogen, Carlsbad, CA, USA) was used to extract total RNA. The RNA quality was measured by an Agilent 2100 Bioanalyzer (Agilent Technologies, Santa Clara, CA, USA). The RNA-seq libraries were prepared using the TruSeqTM RNA sample preparation kit (Illumina, San Diego, CA, USA). We used PCR to complete the entire library preparation, and the integrated library was performed using Illumina HiSeqTM. The sequencing platform at Guangzhou Kidio Biotechnology Co., Ltd. (Guangzhou, China) verified that the selected reference genome of *V. alginolyticus* (accession number: NCBI GCF _ 000354175.2) can be used for further data analysis.

We used PCR to complete the entire library preparation, and the integrated library was performed using Illumina HiSeqTM. The sequencing platform at Guangzhou Kidio Biotechnology Co., Ltd. (Guangzhou, China) verified that the selected reference genome of *V. alginolyticus* (accession number: NCBI GCF _ 000354175.2) can be used for further data analysis. For the annotation of mRNAs, the clean unigenes were compared with different databases, including NCBI NR protein, STRING, SWISS-PROT, and Kyoto Encyclopedia of Genes and Genomes (KEGG), using BLASTX to identify the proteins that shared the highest sequence similarity with the identified unigenes.

The virulence gene sequences of *pstS* mutant strains were amplified by the NCBI database and biological software Clustalx 1.8, and the amino acid sequence alignment and homology analysis were carried out. A phylogenetic tree was constructed with the neighbor-joining method using MEGA7.0 [25].

### 2.12. Prediction of Protein Secondary Structure Models

SOPMA was used for calculating the secondary structural features of the selected protein sequences. We compared the secondary structure prediction by using the Protean of DNAstar lasergene software, the hydrophilicity of PstS was predicted by the Kyte–Doolittle plotting method [26], and the theoretical antigen epitopes of the PstS was predicted by the Jomeson–Wolf method [27].

### 2.13. Statistical Analysis

After acquiring uniquely mapped read counts, we used the package edgeR (targeting mRNAs of the host) to confirm DEGs. The DEGs of EG and CG samples were detected using the DESeq2 (http://bioconductor.org/packages/stats/bioc/DESeq2/, accessed on 14 December 2020) in order to detect the differential genes between the two samples and use FDR < 0.05 & |log_2_FC| ≥ 1 and *q* values <0.05 as the screening conditions. The H-cluster method was used to confirm the expression patterns of DEGs through cluster analysis. All DEGs and gene ontology (GO) terms were mapped to a reference database (http://www.geneontology.org/, accessed on 14 December 2020) to indicate gene function in the samples. We used the KOBAS software to conduct an enrichment analysis of DEGs in the Kyoto Encyclopedia of Genes and Genomes (KEGG) pathway enrichment analyses. The calculation was based on Fisher’s exact test, and we analyzed KEGG pathway enrichment. We analyzed KEGG pathway enrichment using the Benjamini–Hochberg False Discovery Rate method. The data of significantly expressed genes were validated using qRT-PCR. The genes and specific primer sequences used here are listed in Table 3. The expression quantitative software RSEM quantitatively analyzes the gene expression level, calculates the correlation coefficient between each sample, and ensures the rationality of the experimental design. The data included three test samples and three standard samples, as above.

### 2.14. Data Access

The RNA-seq data were deposited in the GenBank SRA database under accession numbers SRS9050393 (CK _ 1 strain group), SRS9050395 (CK _ 2 strain group), SRS9050394 (CK _ 3 strain group), SRS9050397 (*pstS* _ RNAi _ 1 strain group), SRS9050398 (*pstS* _ RNAi _ 2 strain group), and SRS9050399 (*pstS* _ RNAi _ 3 strain group).

## 3. Results

### 3.1. Isolation and Identification of Mutants

Stable silencing strains *pstS*-RNAi-217, *pstS*-RNAi-571, *pstB*-RNAi-33 and *pstB*-RNAi-216 were constructed, and the genes expression levels of the stably silenced strains *pstS*-RNAi-217, *pstS*-RNAi-571, *pstB*-RNAi-33, and *pstB*-RNAi-216 were reduced by 97.8%, 63.1% (Figure 1A), 89.2%, and 62.7% (Figure 1B), respectively, compared to that of the control *V. alginolyticus* ND-01 gene (** *p* < 0.01). Therefore, *pstS*-RNAi-217 and *pstB*-RNAi-33 strains were selected for further study and named *pstS*-RNAi and *pstB*-RNAi.

### 3.2. Biological Characteristics of pstS-RNAi and pstB-RNAi Strains

Our observations suggested that the growth curve could be consequently decreased by stably silencing the genes *pstS* and *pstB*. The growth curve of the *pstS*-RNAi strain, *pstB*-RNAi strain, and *V. alginolyticus* ND-01 strain showed (Figure 1C) that the growth of the stable silencing strains *pstS*-RNAi and *pstB*-RNAi was better than that of wild strain in the early stage of the bacterial growth period. After the stationary phase of bacterial growth, the growth of stable silencing strains was significantly slower than that of wild strains during the later growth stage of bacteria.

The adhesion ability of *V. alginolyticus* control strain (ND-01) and *pstS*-RNAi and *pstB*-RNAi stable silencing strains is shown in Figure 2A. The numbers of the adherent bacteria of control *V. alginolyticus* ND-01 were 1728 ± 27/field of view while those of the strains of *pstS*-RNAi and *pstB*-RNAi were 411 ± 72, and 447 ± 70/field of view, respectively.

A comparison between the abilities of the control *V. alginolyticus* ND-01 and *pstS*-RNAi and *pstB*-RNAi stable silencing strains to form bacterial biofilms is shown in Figure 2B. Compared to that of the control wild *V. alginolyticus* strains, the biofilm formation ability of stable silencing strains *pstS*-RNAi and *pstB*-RNAi decreased significantly by 68.13% and 55.64%, respectively. That reveals the results that *pstS* and *pstB* are closely related to the biofilm formation ability of *V. alginolyticus* and promote biofilm formation by affecting adhesion. We observed that the extent of chemotaxis towards the skin mucus of *E. coioides* was higher in the control *V. alginolyticus*, and the motility results showed that the wild strains formed large round colonies on the semi-solid agar plate. In contrast, the movement ability of stably silenced strains (*pstS*-RNAi) was significantly weakened.

The stable silencing of the *pstS* genes could significantly reduce the chemotactic ability of bacteria, which had decreased by 33.33%, respectively, compared to that of the control *V. alginolyticus* strain (Figure 2C,D). However, we found that the chemotaxis ability of the mucus of *E. coioides* of stably silenced strains (*pstB*-RNAi) had not decreased significantly, and *pstB* was unrelated to bacterial chemotaxis.

### 3.3. Amino Acid Sequence Homology Analysis of Virulence Genes

To construct the phylogenetic tree of PstS, we selected 11 sequences of *Vibrio*, *Streptomyces,* and *Pseudomonas* using the Neighbor-Joining method. We studied the specificity of the phosphate transport system between species, and the amino acid sequence of the *pstS* virulence gene fragment was cloned and determined. The results of the multiple sequence alignment showed that the PstS in *V. alginolyticus* (ND-01) was the closest to the protein of *V. alginolyticus* NBRC 15630 ATCC 17749 in the database and was clustered into one branch (Figure 3). The amino acid sequence of similar genes to *Edwardsiella ictaluri* (strain: S07-698) and *Escherichia coli* str. K-12 substrate MG1655 moves farther and farther away.

### 3.4. Prediction of Protein Secondary Structure Models

In our model, the prediction of protein secondary structure of *V. alginolyticus* PstS by SOMPA shows that the alpha helix (Hh) contains 108 amino acids, accounting for 39.56%; extended strand (Ee) contains 57 amino acids, accounting for 20.88%; beta-turn (Tt) contains 24 amino acids, accounting for 8.79%; random coil (Cc) contains 84 amino acids, accounting for 30.77% (Figure 4A). The results show that more than 55% of the hydrophilic regions of PstS are positive (Figure 4B), indicating that PstS is hydrophilic and belonged to water-soluble proteins. The immunogenicity of the selected PstS sequence fragment is directly predicted by the Jomeson–Wolf method. The results show that the PstS fragment contain 11 antigenic epitopes at 21–27, 45–54, 58–64, 79–92, 97–103, 107–113, 126–140, 146–153, 226–231, 241–248, and 260–268 (Figure 4C).

### 3.5. Effect of pstS Gene RANi on RNA-seq Sequence Data

Transcriptome data analysis showed that the expression levels of *pstS* and *pstB* in phosphate-specific transport systems were significantly reduced by 2.44- and 1.4-fold (** *p* < 0.01). We used qRT-PCR to validate the RNA-seq results, and the relative expression levels of *pstS* and *pstB* genes were 2.9- and 1.77-fold, respectively (Figure 5). Thr results of qRT-PCR were consistent with the trend of RNA-seq analyses, which verified the reliability and accuracy of RNA-seq data.

RNA sequences of the *pstS*-RNAi strain and the *V. alginolyticus* ND-01 strain were analyzed. The number of unknown bases is less than 10% (the results are shown in Appendix A (Figure A1). The base Q20 (percentage of bases with quality value ≧ 20) of each sample is more than 98%, so the quality of the sequence data meets the requirements of subsequent analysis (Table 4). The base error rates of all samples were less than 1% (the results are shown in Appendix A Figure A2). We compared the trimmed sequences with the reference genome (Table 5), and the coverage of the sequences of wild *V. alginolyticus* ND-01 and stable strains was over 96%. The principal component analysis (PCA) revealed the clear segregation of samples (shown in Appendix A Figure A3). EdgeR was used to calculate the DEGs, and we identified 2189 DEGs from the *pstS*-RNAi strains (Figure 6A) and compared to those of wild *V. alginolyticus* ND-01, we observed that 1055 and 1134 genes were significantly up-regulated and down-regulated, respectively (Figure 6B). Among the up-regulated genes, N646_RS04260 (log_2_FC = 7.30) had the largest multiple changes. Among the down-regulated genes, N646_ RS02935 (log_2_FC = −15.95) had the largest change in multiples. The expressions of many genes had changed in *pstS*-RNAi, which suggested that *pstS* acted to regulate the genes in *V. alginolyticus*. Compared with the wild strain of *V. alginolyticus* (ND-01), a gene ontology (GO) analysis of DEMs in *pstS*-RNAi strain was conducted to identify the functions of genes (Figure 6A,B).

According to log_2_FC, the top 30 up-regulations and down-regulations of differentially expressed mRNAs (DEMs) are shown in Figure 6C [28]. We used qRT-PCR to analyze the randomly selected ten up-regulated and ten down-regulated DEMs, and these results of RT-qPCR and the RNA-seq data are consistent and can support the accuracy of the RNA-seq results (Figure 6D).

Compared to those in the wild strain of *V. alginolyticus* (ND-01), the DEMs were found in *pstS*-RNAi strain, including twenty GO secondary terms for seven biological processes, two cell components, and eleven molecular functions, respectively (Figure 7). These genes are mainly involved in biological regulation (122 genes, 5.57%), cell process (662 genes, 30.24%), localization (161 genes, 7.35%), metabolic process (534 genes, 24.39%), cell anatomical entities (830 genes, 37.91%), binding (681 genes, 31.11%), catalytic activity (942 genes, 43.03%), and transportation activities (173 genes, 7.9%).

EdgeR was used to calculate the DEGs between the wild strain of *V. alginolyticus* (ND-01) and *pstS*-RNAi strain, all DEGs in *V. alginolyticus* were mapped to the reference KEGG pathways. The main pathways mapped included those for the “ribosome synthesis-related processes”, “organelles and biosynthesis”, and “pathogenesis and bacterial secretion.” On comparing the expressions of the DEGs involved in these pathways, the largest number of pathways mapped was the “cellular process.” The results of KEGG enrichment analysis is shown in Figure 8. According to the KEGG database, we mapped 40 secondary pathways, many mapped included those for the “amino acid metabolism”, “carbohydrate metabolism”, “energy metabolism”, “cofactors and vitamin metabolism”, “membrane transport”, and “signal transduction and cell community-prokaryote secondary pathways.”

By using Fisher’s exact test, according to the KEGG database, all DEGs in *V. alginolyticus* were mapped to 11 metabolic-related KEGG pathways, the pathways mapped included those for the “namely ribosome”, “C5-branched dicarboxylic acid metabolism”, and “valine, leucine and isoleucine biosynthesis.” We also mapped the “bacterial secretion system”, which was found to be involved in the virulence-related KEGG pathway. According to the GO functional enrichment diagram (Figure 9A), all DEGs in *V. alginolyticus* were mapped to the pathogenesis and bacterial secretion, which were related to bacterial virulence. The pathways mapped included those for the “protein transmembrane transport”, “protein transport”, “protein localization”, “protein transmembrane extracellular transport”, “secretion”, “protein secretion”, “cell secretion”, “sec complex transport protein”, “establishment of protein localization”, and “organic transport.” To reveal the molecular mechanism of the *pstS* gene involved in regulating the virulence of *V. alginolyticus*. The KEGG pathway enrichment diagram (Figure 9B) shows that the DEGs are significantly enriched in the bacterial secretion system, which is closely related to bacterial virulence. These genes were mainly involved in the “bacterial secretion system”, “protein secretion”, “biofilm formation—*V. cholerae*”, “biofilm formation—*Pseudomonas aeruginosa*”, and “quorum sensing and *Yersinia* infection.” These differential genes were associated with target genes for the *pstS* gene to participate in the regulation of *V. alginolyticus* virulence.

## 4. Discussion

*V. alginolyticus* is responsible for causing significant diseases and huge economic losses in cultured fish. Therefore, understanding the virulence mechanism of *V. alginolyticus* is important. Recent research has found that *t**oxR* may be involved in the early stages of the infection of the intestine by enhancing the biofilm information of *V. alginolyticus* via the modulation the expression of glutamine synthesize, levansucrase, and OmpU [29]. Our previous studies indicated that *F. schisandrae* played an anti-*V. alginolyticus* effect in both in vitro and in vivo experiments [7]. Here, the RNA-seq virulence identified the mechanisms of *V. alginolyticus*-resistant *F. schisandrae*, revealing that the resistance system of *V. alginolyticus* and the expression levels of *pstS* and *pstB* genes in the Pst system were significantly down-regulated. The results in this study also indicated that the *pstS* and *pstB* gene were bacterial virulence factors of *V. alginolyticusis*. Bacterial adhesion is an important virulence factor of *V. alginolyticus*. The ability of the pathogens to adhere to the mucus of fish was related to causing infectious diseases [30]. PstS-1 is an adhesin that can promote the phagocytosis of *Mycobacterium tuberculosis*, which is a critical virulence factor [31]. The expression of the main adhesin was significantly reduced and the adhesion to epithelial cells was also reduced considerably in the *pstS* operon mutation in enteropathogenic *E. coli* [32]. However, the virulence mechanisms of *pstS* and *pstB* genes in the Pst system have not yet been completely elucidated. Our study confirmed that the adhesion ability of *V. alginolyticus* was significantly reduced after the stable silencing of the genes in the oxidative phosphorylation pathway [33]. The bacterial adhesion of *V. alginolyticus* significantly decreased upon RNAi-mediated gene silencing, indicating that *pstS* plays an important role in this process.

Biofilm formation is a complex multi-step process and causes diseases [29]. The Pst system-related gene mutations decrease the extent of biofilm formation in *P**. aeruginosa*, *Proteus mirabilis*, *Vibrio cholera*, and *Pseudomonas aureus* [34,35]. Similar to these results, in our study, the biofilm formation ability of *V. alginolyticus* was reduced in stably silenced strains (*pstS*-RNAi and *pstB*-RNAi), indicating that *pstS* and *pstB* genes were related to biofilm formation.

On the other hand, a serious reduction in the motility of *pstS*-RNAi and *pstB*-RNAi may decrease the extent of biofilm formation, as bacterial motility is closely related to its pathogenicity [23]. The Pst system of *P**. aeruginosa* and *A**. baumannii* is correlated with bacterial motility. This study also found that after stable gene silencing, the motility of *V. alginolyticus* was significantly reduced, indicating that *pstS* and *pstB* genes had regulatory effects on its motility [10,36]. Similar to these results, in this study, the motility of *V. alginolyticus* was significantly reduced in *pstS*-RNAi and *pstB*-RNAi, indicating that *pstS* and *pstB* genes had effects on bacterial motility [10,36].

Motility and chemotaxis have long been implicated in bacteria, and chemotaxis appeared to have an enhanced ability for bacteria to invade the host [37]. In recent years, it has been found that *pstS* can perceive the change of inorganic phosphorus concentration in the environment and participate in bacterial chemotaxis [38]. Similar to these results, in our study, the biofilm formation ability of *V. alginolyticus* was reduced in stably silenced strains (*pstS*-RNAi and *pstB*-RNAi), indicating that *pstS* and *pstB* genes were related to biofilm formation. The chemotaxis of *V. alginolyticus* on the mucus of *E. coioides* was significantly reduced in *pstS*-RNAi strains and displayed no change in *pstB*-RNAi strains. The results showed that the *pstS* gene was related to the chemotaxis of *V. alginolyticus*.

Transcriptome sequencing (RNA-seq) is analyzed at the transcriptional level of genes through bioinformatics [39]. Bacterial pathogenicity is mainly mediated by many toxins, we focus on the toxin-mediated pathogenic processes. The bacterial adhesion exhibited significant decreases after the stable gene silencing of *yajC*, which indicated that *yajC* played role in the bacterial adhesion of *V. alginolyticus* [13]. The *Hcp* gene plays an essential role in the biofilm formation of *Shewanella frigidimarina* [40]. After the stable silencing of *the clpV* gene by RNAi technology and the pathogen–host interaction transcriptome sequencing was performed, the flagellar-related genes were significantly down-regulated, indicating that *clpV* was an important bacterial virulence gene [41].

Recent studies have shown that the loss of a single gene can change the landscape of bacterial transcriptomes [42]. *pstS* loss significantly decreased bacterial adherence and invasion into A549 cells and increased A549 cell viability. *pstS* loss also reduced the motility and biofilm-forming ability of *A**. baumannii*, and the minimum lethal dose required by *A. baumannii* ATCC 17978 Δ*pstS* was lower compared to the wild type; thus, the loss of the phosphate sensor *PstS* produced a decrease in *A. baumannii* pathogenesis, supporting its role as a virulence factor [43]. The insertional inactivation of *pstS* has multiple effects on virulence traits, such as increased sensitivity to the bactericidal effect of serum and a significant reduction in the amount of capsular antigen at the cell surface, and the deletion of the *pstS* genes in an ExPEC strain belonging to the avian pathogenic *E. coli* APEC group was shown to reduce virulence in a chicken infection model. The *pst* mutation affects multiple virulence attributes. The phosphate-specific transport (Pst) system is part of a complex network important for both bacterial virulence and the stress response of *A*. *baumannii* [10]. Similar to these results, in our study, the stable silencing of *pstS* had significantly altered the transcriptome of *V. alginolyticus* compared to its wild type, and the result of GO analysis revealed that many DEGs were found to mainly involved in “biological regulation”, “cell process”, “localization”, “metabolic process”, “cell anatomical entities”, “binding”, “catalytic activity”, and “transportation activities.” The result revealed that many DEGs, which were related to bacterial functional categories, were involved in the localization and a response to the stimuli of bacteria [44]. KEGG analysis showed that the silencing of the *pstS* gene had a significant change in the expression of bacterial virulence. Among the 11 metabolic-related KEGG pathways, some were related to virulence, which involved “metabolic pathways”, “ribosomes”, and “bacterial secretion systems.” The “bacterial system” transported secreted proteins to the periplasm via the Sec system and transported proteins to the extracellular by GSP protein complexes [45], and a “bacterial secretion system” was able to mediate adhesion by controlling the secretion of intercellular polysaccharide adhesins. These KEGG analysis results indicated that the DEGs confirmed in our study was able to significantly affect the adhesion of *V. alginolyticus*, which suggests that *V. alginolyticus* might decrease stimulus for the adhesion caused by the stable silencing of *pstS* through the regulation of adhesion-related pathways. In our study, the expression levels of *pstS*-RNAi were significantly down-regulated. The *pstS* mutation affects multiple virulence factors, which revealed the molecular mechanism of the *pstS* gene involved in regulating the virulence of *V. alginolyticus*.

## 5. Conclusions

In this study, our result displayed that the adhesion, biofilm formation ability, and motility of stable silencing strains (*pstS*-RNAi and *pstB*-RNAi) significantly decreased compared with those of the wild-type strain; moreover, the chemotaxis of *pstS*-RNAi was reduced. Therefore, *pstS* and *pstB* could regulate bacterial virulence of *V. alginolyticus* by affecting its adhesion, biofilm formation ability, and motility. Moreover, stable silencing of *pstS* and *pstB* had altered the transcriptional landscape of *V. alginolyticus*, and regulated expressions of the genes involved in virulence-related pathways. Understanding the relationship of the expressions of *pstS* and *pstB* with bacterial virulence could provide better insights into understanding the virulence mechanism of *V. alginolyticusis* and provide new perspectives to prevent bacterial diseases.

## Figures and Tables

**Figure 1 microorganisms-10-02093-f001:**
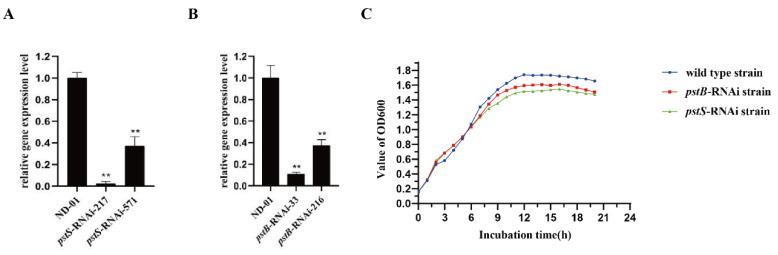
Construction and growth curve of *pstS*-RNAi and *pstB*-RNAi strains. (**A**) The expression level of *pstS* gene in two mutants; (**B**) The expression level of *pstB* gene in two mutants; (**C**) Growth curves of strains ND-01, *pstS*-RNAi, and *pstB*-RNAi, ** *p* < 0.01.

**Figure 2 microorganisms-10-02093-f002:**
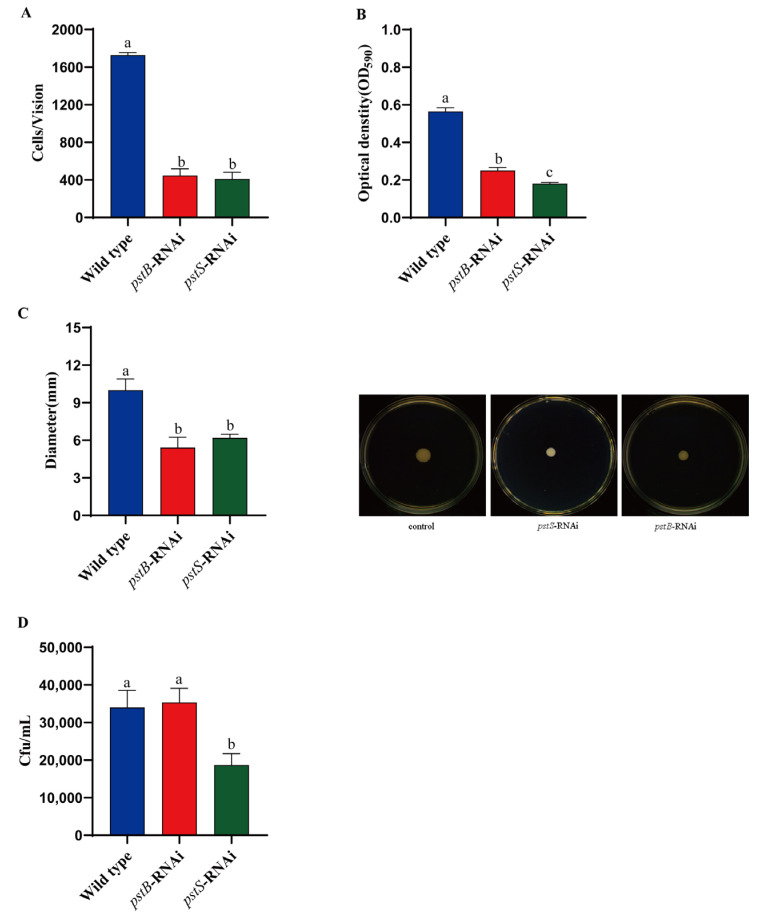
Characteristics of the wild *V. alginolyticus* (ND-01) strain, stable silencing strain *pstS*-RNAi, and *pstB*-RNAi. (**A**) The comparison of the adhesion ability; (**B**) biofilm formation ability; (**C**) motility; and (**D**) chemotaxis of *E**. coioides* mucus were expressed as mean ± SD. Three independent biological replications were performed in each group. Bars with different letters indicate significant differences (*p* < 0.05).

**Figure 3 microorganisms-10-02093-f003:**
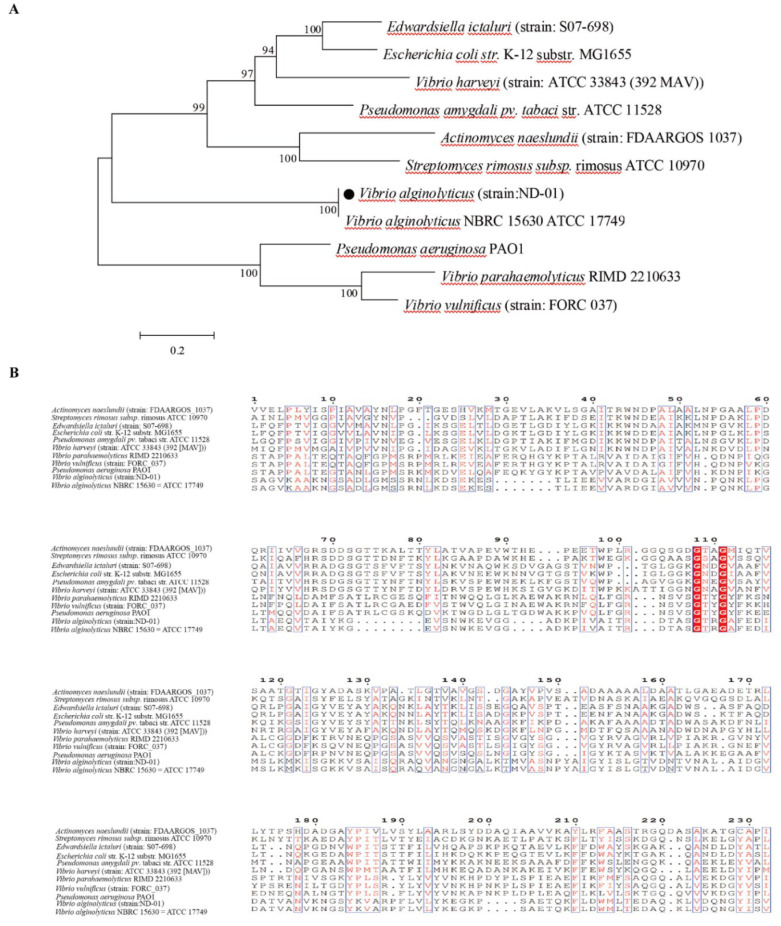
Phylogenetic tree of PstS amino acid sequence of phosphate transport system. (**A**) The phylogenetic tree of PstS; (**B**) Homology Analysis of PstS.

**Figure 4 microorganisms-10-02093-f004:**
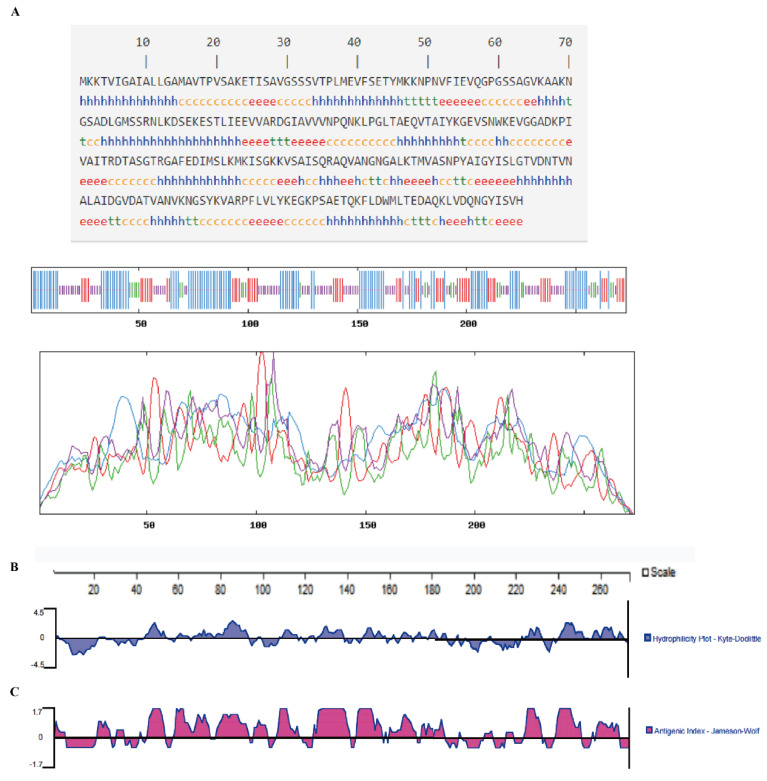
(**A**) Second structure prediction of PstS. h is alpha helix, t is β-turn, e is extended chain, and c is random coil; (**B**) Hydrophilicity analysis of PstS (Hydrophilicity, Kyte–Doolittle Plot); (**C**) Immunogenicity analysis of PstS (Antigenic Index, Jameson–Wolf).

**Figure 5 microorganisms-10-02093-f005:**
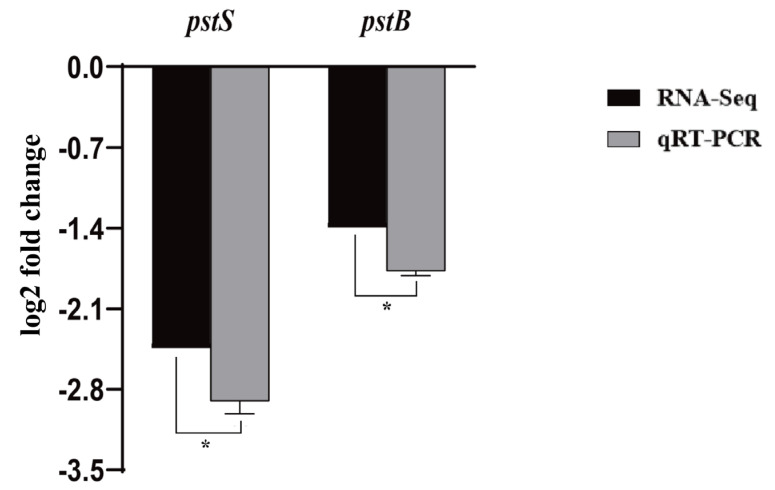
Comparison of transcriptome data and qRT-PCR results: the RNA-Seq data column color is white, the qRT-PCR data column color is black, * *p* < 0.05.

**Figure 6 microorganisms-10-02093-f006:**
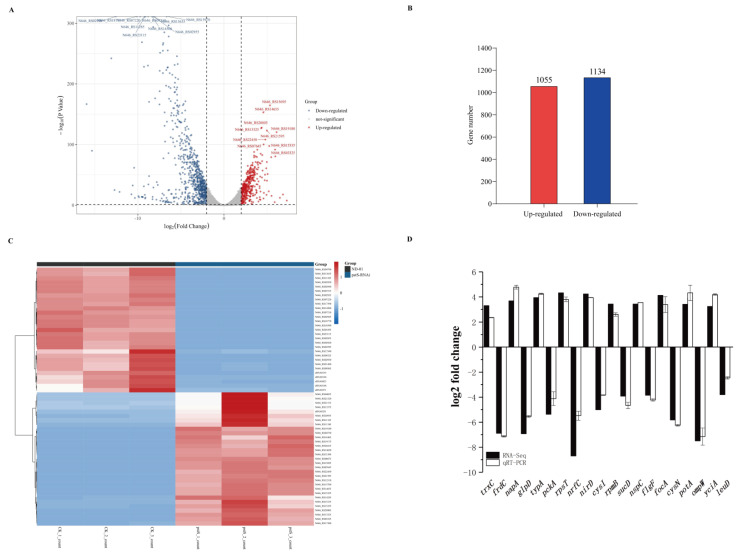
Comparative transcriptome analysis of *V. alginolyticus*. (**A**) Volcano maps obtained by the Edger analysis of *V. alginolyticus* RNA pools (*pstS*-RNAi strain/ND-01 strain); (**B**) Number of up-regulated and down-regulated genes of differential genes; (**C**) Thermal maps of the top 30 mRNA up-regulations and down-regulations in the transcriptome; (**D**) Transcriptome validation of the *pstS*-RNAi strain.

**Figure 7 microorganisms-10-02093-f007:**
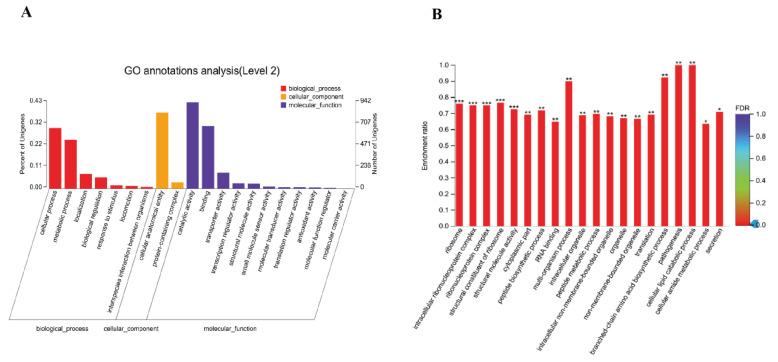
(**A**) GO annotation diagram of differential gene of *V. alginolyticus* stable silencing strain; (**B**) GO enrichment column diagram. * *p* < 0.05, ** *p* < 0.01, *** *p* < 0.001.

**Figure 8 microorganisms-10-02093-f008:**
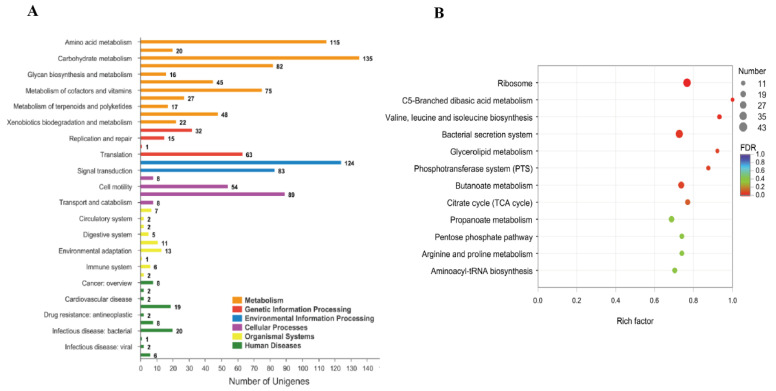
(**A**) KEGG annotation map of differential gene of *V. alginolyticus* stable silencing strain; (**B**) KEGG enrichment bubble diagram.

**Figure 9 microorganisms-10-02093-f009:**
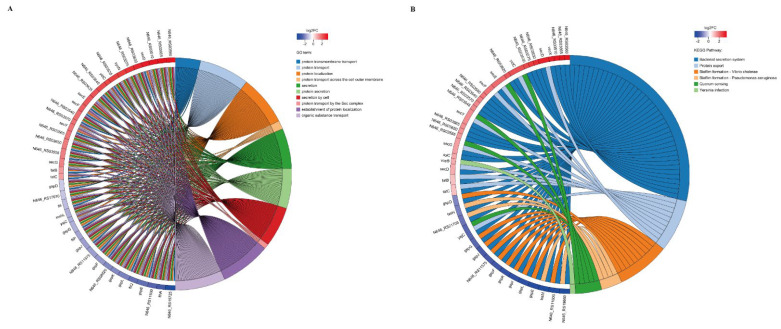
(**A**) GO function enrichment diagram of the differential gene of *V. alginolyticus*; (**B**) Enrichment chord map of the differential gene KEGG pathway in *V. alginolyticus*.

**Table 1 microorganisms-10-02093-t001:** Specific shRNA sequences of *pstS* and *pstB* genes of *V. alginolyticus*.

shRNA	Sequences (5′–3′)
	R:5′ -CAAAAAAGCACAGTTAAGTGGTTCAACGTCTCTTGAACGTTGAACCACTTAACTGTGCG-3′
ShRNA-*pstS*_217_	F:5′ -GATCCGCAGACCTTGGTATGTCTTCTTTCAAGAGAAGAAGACATACCAAGGTCTGCTTTTTTGCATG-3′
	R:5′ -CAAAAAAGCAGACCTTGGTATGTCTTCTTCTCTTGAAAGAAGACATACCAAGGTCTGCG-3′
ShRNA-*pstS*_571_	F:5′ -GATCCGCATCTAACCCATACGCAATCTTCAAGAGAGATTGCGTATGGGTTAGATGCTTTTTTGCATG-3′
	R:5′ -CAAAAAAGCATCTAACCCATACGCAATCTCTCTTGAAGATT GCGTATGGGTTAGATGCG-3′
ShRNA-*pstB*_33_	F:5′ -GATCCGCCACCATTGGACGTTCATAATTCAAGAGATTATGAACGTCCAATGGTGGCTTTTTTGCATG-3′
	R:5′ -CAAAAAAGCCACCATTGGACGTTCATAATCTCTTGAATTATGAACGTCCAATGGTGGCG-3′
ShRNA-*pstB*_216_	F:5′ -GATCCGCGTCTACATGGCAAGAATGTTTCAAGAGAACATTCTTGCCATGTAGACGCTTTTTTGCATG-3′
	R:5′ -CAAAAAAGCGTCTACATGGCAAGAATGTTCTCTTGAAACATTCTTGCCATGTAGACGCG-3′

**Table 2 microorganisms-10-02093-t002:** Primer sequences of *pstS*, *pstB* and 16S of *V. alginolyticus*.

Target Gene	Primers Sequence for Gene Overexpression
*16S*-R	5′-CCTACGGGAGGCAGCAG-3′
*16S*-F	5′-ATTACCGCGGCTGCTGG-3′
*pstS-R*	5′-GAAATGTAGCCGATTGCGTAT-3′
*pstS-F*	5′-AGGCGAAGTATCAAACTGGAA-3′
*pstB-R*	5′-TGACTCGACACCCTTCCACTA-3′
*pstB-F*	5′-AACGGATGAACAAACAGCAAT-3′

**Table 3 microorganisms-10-02093-t003:** Primers for validating transcriptome differential genes by qRT-PCR.

Primer Name	Sequences (5′–3′)
*trxC*-R	5′-AAGTGATACGCCTGTTGTCG-3′
*trxC*-F	5′-CCATGATGGTCGGAATGCT-3′
*frdC*-R	5′-CATCCTTTCTACCGCTTCT-3′
*frdC*-F	5′-ATGTTGATTGCGACTACGA-3′
*napA*-R	5′-AGCAATACGCCAATCCAAAT-3′
*napA*-F	5′-TGGTTGACCCGTTAGTGAGA-3′
*glpD*-R	5′-AGCGGTGTTCGTCCTTTAT-3′
*glpD*-F	5′-CGTCCAGTCTTTACCCATTT-3′
*typA*-R	5′-GGCACAGTAATGGGCTACCT-3′
*typA*-F	5′-CCTTCTTTACCAGCGAATGG-3′
*pckA*-R	5′-CGTAGCGGTATTCTTTGGTC-3′
*pckA*-F	5′-GCTTCTTTCGATAGCTTGATG-3′
*rpsT*-F	5′-TGAGAAACGTCGTCAGCACA-3′
*rpsT*-R	5′-CAGCGAAACGAGACTTATGG-3′
*nrfC*-R	5′-TAAATAAAGTCCCTGAAGGC-3′
*nrfC*-F	5′-CAGCAAGGCAATAACCAC-3′
*nirD*-R	5′-TGTCGCGTTATTCTACATTCC-3′
*nirD*-F	5′-TCTTCCAAACACTGCCCACT-3′
*cysI*-R	5′-TCAGTGAACACCTGCTTCC-3′
*cysI*-F	5′-AACCAACTTGCCGTTATCC-3′
*rpmB*-R	5′-AAGCGTCCAGTAACGGGTAA-3′
*rpmB*-F	5′-CACGCATGTCTGCAAGAACA-3′
*sucD*-R	5′-CAATCCCAGGTTCAAACTTCA-3′
*sucD*-F	5′-ATACGTTTACCAGGAGGAGCA-3′
*nspC*-R	5′-GAGAAACACGGCGTACAAAT-3′
*nspC*-F	5′-AAACAAGAGCACGATCCAAT-3′
*flgF*-R	5′-ACCAACTCTGCCTCAACGC-3′
*flgF*-F	5′-CCAAGAACAGGGAAACCATT-3′
*focA*-R	5′-TGTTCGTGTCGGCAGGTT-3′
*focA*-F	5′-ACCGCCACCAATGATGTT-3′
*cysN*-R	5′-CGTTAATCGTCCTAACCTCG-3′
*cysN*-F	5′-TCATCGCTTAGCGTCAGAGT-3′

**Table 4 microorganisms-10-02093-t004:** Distribution of base mass in *V. alginolyticus* transcriptome.

Sample	Clean Reads Total (Article)	Clean Bases (bp)	Clean Q20 (%)
CK-1	31,287,296	4,082,632,980	98.8
CK-2	27,625,534	3,622,870,442	98.53
CK-3	32,966,152	4,384,526,078	98.77
*pstS*-RNAi-1	24,686,362	3,446,598,515	98.57
*pstS* -RNAi-2	32,052,740	4,492,123,947	98.54
*pstS* -RNAi-3	26,844,948	3,765,247,071	98.37

**Table 5 microorganisms-10-02093-t005:** Mapping ratio statistical results of sequencing data compared with the *V. alginolyticus* reference genome.

Sample	Clean Reads Total (Article)	Total Number of Clean Reads Aligned to the Genome (Article)	Percentage (%)
CK-1	31,287,296	30,800,326	98.44
CK-2	27,625,534	27,002,352	97.74
CK-3	32,966,152	32,531,309	98.68
*pstS*-RNAi-1	24,686,362	23,869,166	96.69
*pstS* -RNAi-2	32,052,740	30,971,033	96.63
*pstS* -RNAi-3	26,844,948	25,876,114	96.39

## Data Availability

Not applicable.

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
