# Peer review of "Mechanisms Underlying the Virulence Regulation of Vibrio alginolyticus ND-01 pstS and pstB with a Transcriptomic Analysis"

_microorganisms, 2022, doi:10.3390/microorganisms10112093_

Round 1

Reviewer 1 Report

I appreciate the Authors’ research work, with practical importance. Their complex experiments definitely proved the roles of pstS and pstB genes and, thus, shed more light on the virulence mechanisms of Vibrio alginolyticus. The important work of this team of researchers resulted in a good manuscript and their results were illustrated in figures and tables. The authors also paid attention to details. I only have some minor comments/suggestions.

1. Abstract: Please define abbreviation “F. (from Fructus) schisandrae” before using it. Also, please re-write the Abstract, mentioning clearly the aims of your current research and your findings, as a synthesis of the whole manuscript. After reading the full manuscript, I understand the abstract. However, in the present form, the Abstract does not do any favor to the manuscript. The Abstract has to be excellent, so that readers will find it interesting enough to read the whole paper and maybe use it in their research, cite it etc.

2. Keywords: It would be advisable to use other Key words, not those belonging to the title. This would increase the likelihood of the paper being found by readers. The importance of Keywords is to improve indexing. Only “Toxicity” is not included in the title.

3. Discussion: Line 410: please clarify the fragment of sentence “the studies which, mainly focus on toxR Gene [29]”. I also suggest inserting the strengths and limitations of your research, as well as some proper directions for future studies.

4. Conclusion: I suggest removing the numbers (1, 2, 3 and 4). Otherwise, it is clear and crispy.

5. Figures: Figure legend 2 - Please mention the exact level of statistical significance for letters a, b and c (p value). Figure 3 – Could you please make it clearer and legible (the amino acid sequence)? Same for Figure 6, A and C? And 9 A and B? So that they are readable and useful.

6. References: Please write all data for refs. 3, 7 and 11. All references should be written in a uniform manner.

Thank you!

Author Response

  We are very grateful for the reviewer’s helpful and thoughtful comments. According to your comments, the presentation style has been improved, the incorrect statements have been revised. The grammatical errors of our manuscript have been examined and improved. The revision made to the manuscript has been marked up using the “Track Changes” function, such that changes can be easily viewed by the editors and reviewers. All changes made to the manuscript are in red so that they can be easily identified in another revision. We hope that these revisions will meet your requests.

Reviewer 2 Report

This is an interesting manuscript, but you may improve this article to publish in this journal. Otherwise, I have a lot of recommendations to increase the quality of your paper. Be careful with the writing and mistakes.

In the article title, you must write the author of the scientific name.

In the Abstract, there are some mistakes.

Line 24. When you write “F. schisandrae” you must write the genus because the abstract is the first contact and public reading of everyone who wants to access to your paper, so you must think about the visibility of your very interesting research. Therefore, you must write the full scientific name, which is “Fructus schisandrae”. In addition, because this is a scientific journal you must write as well the authors of the species.

In the same line, you must write the units in the right way because “ml” is not correct, so, the right way is “mL”. Therefore, you must fix this mistake in the whole paper.

As well in the same line 24 you must written “V. alginolyticus” in a size bigger than the rest of the text. This is a very common mistake in the whole paper please fix this.

Line 36. Please, write the letters that you use for the acronym in capitals. This makes the reading easier. Therefore, you must write “Differentially Expressed Genes (DEGs)”.

When you write an acronym, you must write in brackets its meaning.

There are several keywords repeated in the article title. The keywords are “Vibrio alginolyticus”, “pstS” and “pstB”. In order to increase the visibility of your paper I recommend changing these keywords by others. If you change them by other keyword, you will increase the probability that your paper could be found by future readers when they look for your paper in some databases like Scopus for example. If you repeat the same words in the article title and in keywords, less people could find your work. So, you must think about the visibility of your research. For example, when you look for a specific word or a group of words, normally in the same box you look for “Article title, Abstract and Keywords” at the same time, so, if you have different ones in all of them, obviously the visibility of your manuscript will increase.

Line 61. You must write the family of the species.

In the same line, you must write the author of the species Fructus schisandrae at least the first time that you write it. This is a very common mistake in the whole manuscript please fix it.

Line 64. You must write the author of the species Staphylococcus aureus at least the first time that you write it. This is a very common mistake in the whole manuscript please fix it.

Line 72. You must write the author of the species Epinephelus coioides at least the first time that you write it. This is a very common mistake in the whole manuscript please fix it.

Line 77. You must write the author of the species Acinetobacter baumannii at least the first time that you write it. This is a very common mistake in the whole manuscript please fix it.

Line 112. You must write the units in the right way because “ml” is not correct, so, the right way is “mL”. Therefore, you must fix this mistake in the whole paper.

Line 115. You must write the units in the right way because “ml” is not correct, so, the right way is “mL”. Therefore, you must fix this mistake in the whole paper.

Line 127. You must write the units in the right way because “ml” is not correct, so, the right way is “mL”. Therefore, you must fix this mistake in the whole paper.

Line 351. The letters of the Figure are very tiny. You must increase its size in order to read them.

Even, you cannot read the legend of the Figure 6C, is impossible.

It is very difficult to read the letters of the Figure 9B because they are very tiny.

Line 419. You must write the author of the species Mycobacterium tuberculosis at least the first time that you write it. This is a very common mistake in the whole manuscript please fix it.

Line 429. You must write the author of the species Pseudomonas aeruginosa at least the first time that you write it. This is a very common mistake in the whole manuscript please fix it.

Line 429. You must write the author of the species Pseudomonas aureus at least the first time that you write it. This is a very common mistake in the whole manuscript please fix it.

Line 456. You must write the author of the species Shewanella frigidimarina at least the first time that you write it. This is a very common mistake in the whole manuscript please fix it.

Line 456. You must write the author of the species Bacillus subtilis at least the first time that you write it. This is a very common mistake in the whole manuscript please fix it.

Otherwise, the authors adequately developed the Introduction, presenting the problems.

The methods are adequate.

The Discussion is well developed, and the data presented are correctly compared with other papers.

The authors are to be congratulated for the results obtained in this article.

Author Response

(The authors gave the same response as above.)
